# Born-like limit scattering and pair-breaking crossover in the nodal superconductivity of $(TMTSF)_2ClO_4$

Shota Yano,[1] Kazumi Fukushima,[2] Katsuki Kinjo,[2, 3] Soichiro Yamane,[2, 4] Le Hong Hoàng To,[5] Pascale Senzier,[6] Cécile Mézière,[7] Shamashis Sengupta,[5] Claire A Marrache-Kikuchi,[5, *] Denis Jerome,[6, †] and Shingo Yonezawa[1, 2, 4, ‡]

[1]*School of Science, Faculty of Science, Kyoto University, Kyoto 606-8502, Japan*
[2]*Department of Physics, Graduate School of Science, Kyoto University, Kyoto 606-8502, Japan*
[3]*Institute of Multidisciplinary Research for Advanced Materials, Tohoku University, Sendai 980-8577, Japan*
[4]*Department of Electronic Science and Engineering,*
*Graduate School of Engineering, Kyoto University, Kyoto 615-8510, Japan*
[5]*Université Paris-Saclay, CNRS, IJCLab, 91405, Orsay, France.*
[6]*Université Paris-Saclay, CNRS, Laboratoire de Physique des Solides, 91405, Orsay, France.*
[7]*Univ. Angers, CNRS, MOLTECH-Anjou, SFR MATRIX, F-49000 Angers, France.*
(Dated: January 30, 2026)

In the quasi-one-dimensional organic unconventional superconductor $(TMTSF)_2ClO_4$, the randomness of the non-centrosymmetric $ClO_4$ anions can be experimentally controlled by adjusting the cooling rate through the anion-ordering temperature. This feature provides a unique opportunity to study disorder effects on unconventional superconductivity in great detail. We here report on measurements of the electronic specific heat of this system, performed under various cooling rates. The evolution of the residual density of states indicates that the $ClO_4$ randomness works as Born-limit pair breakers. Furthermore, detailed analyses suggest a peculiar crossover from strong unitarity scattering due to molecular defects toward the Born-limit weak scattering due to borders of ordered regions. This work supports the nodal $d$-wave-like nature of pairing in $(TMTSF)_2ClO_4$ and intends to provide an experimental basis for further developments of pair-breaking theories of unconventional superconductors where multiple electron scattering mechanisms coexist.

## I. INTRODUCTION

The discovery in 1980 of a quasi-one-dimensional (Q1D) organic superconductor $(TMTSF)_2ClO_4$ at ambient pressure by Bechgaard *et al.*[1] has facilitated in-depth studies of its electronic properties, compared to those carried out on other organic superconductors such as the first discovered organic superconductor $(TMTSF)_2PF_6$, for which high pressure is a prerequisite[2–4]. After extensive theoretical and experimental studies, different pairing mechanisms have been proposed to explain the emergence of superconductivity in Q1D organic compounds[5–8], as reviewed in a recent paper[3]. While the issue remains unresolved, one of the most plausible scenarios suggests nodal $d$-wave-like pairing driven by magnetic fluctuations[9–13], as supported by various experiments[14] including NMR $1/T_1$ and Knight shift measurements[15] and angle-dependent magneto-resistance and calorimetry experiments[16,17].

What makes $(TMTSF)_2ClO_4$ peculiar compared to $(TMTSF)_2PF_6$ is that the non-centrosymmetric $ClO_4$ anions are positioned at the inversion centers of the crystal. At high temperatures, the $ClO_4$ orientation is random, so that the inversion symmetry of the crystal is preserved, on average. Below the anion-ordering temperature $T_{AO} = 24$ K, at slow cooling, the entropy gain from reduced degrees of freedom eliminates randomness in the $ClO_4$ orientation, inducing an alternating anion ordering along the $b$ axis. Thermodynamic and magnetic investigations have revealed that the nature of the $(TMTSF)_2ClO_4$ ground state is profoundly dependent on the speed at which the sample is cooled down across

$T_{AO}$[18–20]. Fast cooling makes it possible to retain the randomness in the $ClO_4$ orientation down to lowest temperatures, leading to a good nesting of the single-pair Fermi surfaces. This nesting stabilizes the insulating spin density wave (SDW) phase. At slow cooling rates, on the other hand, the alternating orientation of the anions gives rise to a folding of the Fermi surface. Such folding suppresses the SDW phase, enabling superconductivity below $T_c = 1.2$ K.

Recent simultaneous measurements of transport and magnetic properties of $(TMTSF)_2ClO_4$ revealed the existence of a crossover between homogeneous and granular superconductivity when increasing the cooling rate above about 1 K/min across $T_{AO}$[21,22]. For slow cooling rates, nonmagnetic disorder arises from small randomly distributed clusters of disordered anions, which act as scattering centers and lead to the suppression of superconductivity. At cooling rates above 1 K/min, the system behaves as a network of randomly distributed superconducting (anion-ordered) regions embedded within a normal-conducting matrix with disordered anions. Global superconductivity then occurs due to the proximity effect between neighboring superconducting regions at a critical temperature calculated in Ref. 21. Based on these results, we were able to relate the cooling rate to the elastic electron mean free path in this system.

Thus, $(TMTSF)_2ClO_4$ serves as a textbook case of an unconventional superconductor in which it is possible to study in detail the role of disorder on the superconducting ground-state stability.

A basic property of $s$-wave superconductivity proposed

in the BCS theory is the isotropic ($k$-independent) gapping on the Fermi surface. Hence, no pair breaking is expected from the scattering of electrons against spinless impurities[23], since such scattering essentially mixes and averages gaps at different $k$ without any effect on $T_c$. However, this so-called Anderson's theorem is no longer valid in the case of an anisotropic gap with sign changes of the gap. Consequently, $T_c$ for unconventional superconductors should be strongly affected by any nonmagnetic scatterings. Actually, the sensitivity of the superconducting state to nonmagnetic defects has been considered as a solid indication of an unconventional pairing mechanism. The effects of nonmagnetic impurities on $T_c$ in such superconductors have been derived by generalizing the conventional Abrikosov-Gorkov (AG) pair-breaking theory for magnetic impurities to non-$s$-wave superconductors[24]. However, while this extension predicts the effect of increased scattering on $T_c$, it does not provide any information on the character of the dominant scatterers.

Such pair-breaking information can be studied through the evolution of the quasi-particle density of states (DOS). Theoretically, two extreme models for impurity scattering have been proposed, the unitarity limit and Born limit. In the unitarity limit, where randomly distributed strong scatterers are assumed, increase of such scatterers results in a strong renormalization and rapid increase of DOS as the scattering rate increases[25,26]. In contrast, in the Born limit, where weak scatterers are assumed to be distributed uniformly, the residual DOS is hardly affected as the scattering rate increases, at least down to $T_c/T_{c0} \approx 0.5$, where $T_{c0}$ is the critical temperature in the hypothetically perfectly clean limit[27–29]. It is important to note that, for both scattering limits, the dependence of $T_c$ on the scattering rate is expected to follow the same AG curve[30,31]. As a consequence, only the residual DOS, which can be measured by the electronic specific heat, the Knight shift or the nuclear relaxation rate, can distinguish between the two limits.

To the best of our knowledge, a theoretical framework bridging the weak (Born) and strong (unitary) scattering limits remains to be developed, as some compounds exhibit intermediate scattering strengths not fully captured by either model.[32,33] When interpreting data from newly studied materials, the appropriate scattering model is often not known a priori. It is therefore reasonable to begin with the simplest cases, such as the two extreme approximations.

In this paper, we have measured the residual DOS in the low temperature limit, in order to determine the nature of the scattering channel. By measuring the electronic specific heat of a single $(TMTSF)_2ClO_4$ sample with a careful control of the cooling rate across $T_{AO}$, we can control the randomness of $ClO_4$ anions while keeping the amount of chemical defects constant. Interestingly, we reveal that the $ClO_4$ randomness works as Born-limit scatterers. We also address the important issue of the electron scattering strength when supercon-

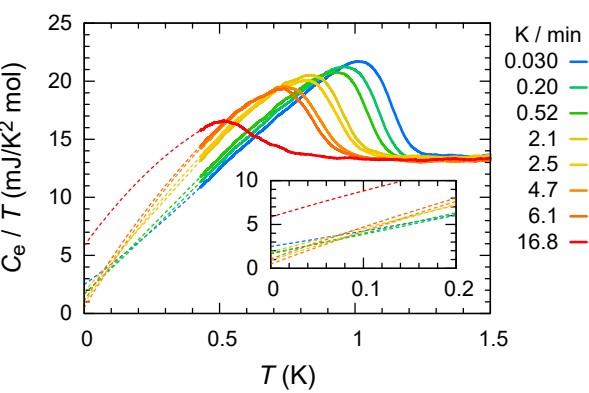

FIG. 1: Temperature dependence of the electronic specific heat $C_e$ of $(TMTSF)_2ClO_4$ divided by temperature $T$ measured after cooling through $T_{AO}$ with different cooling rates. The experimental data are shown in solid curves and the extrapolation to low temperature (see text) with dashed curves. For clarity, the extrapolated curves at very low temperature are shown in the inset. The electronic specific heat of the normal state after subtraction of the phonon contribution amounts to $\gamma_N = 13.3 \pm 0.2$ mJ/K$^2$ mol.

ductivity crosses over from uniform to granular at fast cooling rates.

## II. EXPERIMENTAL SETUP

We performed specific heat measurements on a single crystalline sample of $(TMTSF)_2ClO_4$ (mass of 0.364 mg) prepared by electrocrystallization[34], using a custom-made calorimeter, described in Appendix A, placed into a commercial cryostat (Quantum Design, PPMS) equipped with the adiabatic-demagnetization refrigerator (ADR) option. This setup is capable of cooling a sample well below $T_c$ (see Appendix B), while allowing us to control the cooling rate around $T_{AO} = 24$ K in a wide cooling-rate range[22,35]. With this method, the slowest cooling rate was of 30 mK/min, and the base temperature was 400 mK.

For the calorimetry, we use the AC method[36], in which a sinusoidal current of frequency $\omega_H$ is supplied to the heater and the temperature oscillations induced by Joule heating at the frequency $2\omega_H$ are detected with lock-in amplifiers (Stanford Research Systems, SR830 and SR860). Here we typically choose $\omega_H/2\pi = 21.14$ Hz. From the raw heat capacity data, the contributions of the background due to the experimental setup without the sample and of the phonons were subtracted, using the procedure described in Appendix C, to obtain the temperature dependence of the electronic specific heat $C_e$ of $(TMTSF)_2ClO_4$.

## III.   RESULTS

### A.   Evolution of $T_c$ with disorder

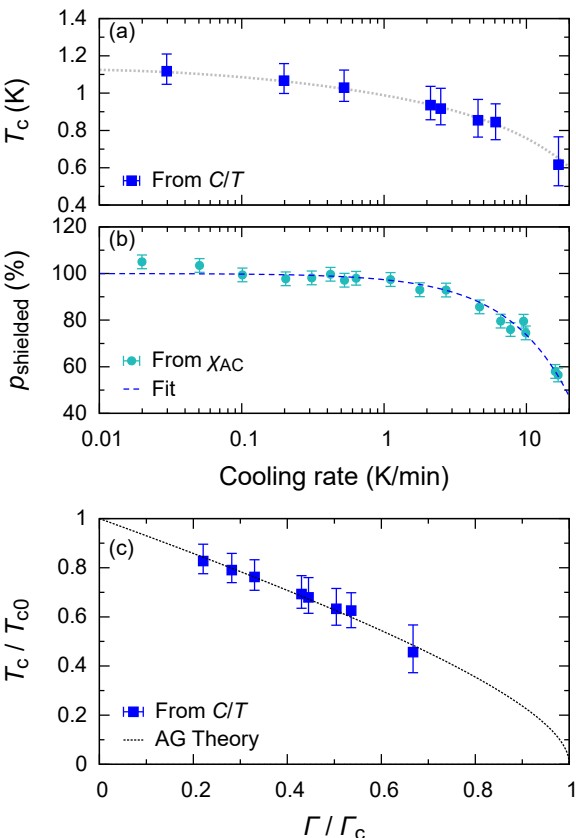

FIG. 2: Cooling rate dependence (a) of the critical temperature $T_c$ (the dotted lines is a guide to the eye) and (b) of the sample shielded volume fraction $p_{shielded}$ (data from Ref 22). The dotted line is a linear fit to the data. (c) Normalized $T_c$ as a function of the normalized scattering rate $\Gamma/\Gamma_c$ with $T_{c0} = 1.35$ K (see text). The scattering rate is deduced from previous resistivity measurements[22]. The dotted curve displays a result of the fitting of the Abrikosov-Gorkov (AG) theory.

In Fig. 1, we plot temperature dependence of $C_e/T$ measured after cooling across $T_{AO}$ with various cooling rates ranging from 0.03 K/min to 16.8 K/min. As the cooling rate increases, superconductivity is clearly suppressed. From this data set, we determined the thermodynamic superconducting critical temperature $T_c$ by a triangular analysis of $C_e/T$ using the usual entropy conservation rule for a broad transition. As expected, $T_c$ decreases when the disorder increases due to a larger cooling rate (Fig. 2(a)).

As mentioned earlier, in $(TMTSF)_2ClO_4$, superconductivity is homogeneous up to a cooling rate of about 1 K/min. For faster cooling rates, randomness makes superconductivity granular in nature, which results in only part of the sample being shielded by superconducting currents. Figure 2(b) shows the sample shielded volume fraction $p_{shielded}$ as measured by AC susceptibility for the same sample.

In our previous work[22], for a $(TMTSF)_2ClO_4$ sample of similar quality[34], we had established the relation between the cooling rate and the normal-state residual resistivity along the $c^*$ axis $\rho_{c^*}$. From this, we can derive the dependence of $T_c$ on the elastic scattering rate $\Gamma$[37], as shown in Fig. 2(c). In this work, we focus on the normalized scattering rate $\Gamma/\Gamma_c$. Detailed discussions on the absolute scattering strength have been already done in the literature[38,39]. This result is well fitted by the Abrikosov-Gorkov theory[31], giving $T_{c0} = 1.35 \pm 0.03$ K in the hypothetical perfectly clean limit and the critical resistivity $\rho_{c*0} = 0.14$ $\Omega$.cm . This value of $T_{c0}$ is consistent with previous results[22,38] and the critical resistivity value agrees reasonably as well. Figure 2(c) tells us that $T_c$ is strongly affected by non-magnetic scattering centers as the cooling rate increases, as expected for unconventional superconductivity. However, it is unable to provide any information on the strength of these scatterers. To obtain such information, one must determine the residual DOS at zero temperature.

### B.   Low-temperature electronic specific heat

Although the lowest system temperature of the ADR is around 0.15 K, the lowest sample temperature achieved was around 0.4 K due to weak thermal coupling. Therefore, we extrapolate $C_e(T)/T$ curves to $T = 0$ in order to obtain the residual $C_e/T$ values. To do so, we used extrapolation functions with three parameters such as $f(T) = a_0 + a_1 T + a_2 T^2$ (see Appendix D for more details). The parameters of such functions can be uniquely determined by considering the three conditions: continuity of the function and of its derivative with the experimental data at a connection temperature $T_0$ in the temperature range of 0.4-0.5 K: $f(T_0) = C_e(T_0)/T_0$ and $df/dT|_{T=T_0} = (d/dT)(C_e/T)|_{T=T_0}$, and the entropy-balance equation

$$\int_0^{T_0} f(T)\,dT + \int_{T_0}^{T_{c,onset}} \frac{C_e}{T}\,dT = \gamma_N T_{c,onset},$$

where $\gamma_N$ is the normal-state electronic specific heat coefficient obtained after subtracting the phonon contribution and $T_{c,onset}$ is the temperature below which $C_e$ departs from its normal state value $\gamma_N T$. This extrapolation provides a good estimation of the residual $C_e/T$ as $\gamma_0 = C_e/T|_{T\to 0} \sim a_0$. The extrapolation functions are shown as dashed curves for $T \lesssim 0.4$ K in Fig. 1, and the residual DOS $\gamma_0$ determined in this manner is shown in Fig. 3(a).

We comment here on various factors that may affect the low-temperature extrapolation. Multiband effects are

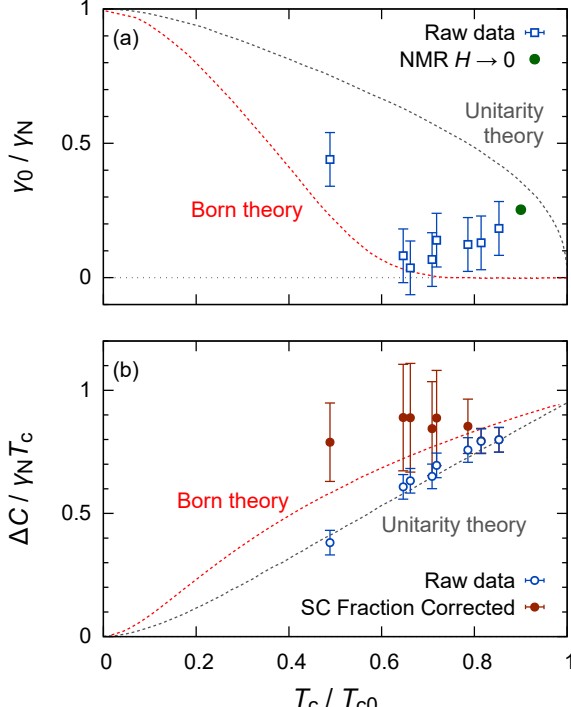

FIG. 3: (a) Normalized values of the residual quasiparticle DOS, obtained from specific heat data, plotted against the normalized $T_c$. Theoretical values for strong scattering (unitarity limit) and weak scattering (Born limit) are also plotted (dotted curves)[30]. The error bars provide an estimate of the uncertainty of the extrapolation procedure. The green data point is extracted from NMR data[15,40] (cooling rate of 7 mK/min, $T_{c0} = 1.4$ K). (b) Normalized specific heat jump at $T_c$ as a function of $T_c$ /$T_{c0}$ for the raw data (error bars correspond to the error in the determination of the jump amplitude) and after correction (error bars comprise the experimental uncertainty and the uncertainty on $p$) for the superconducting (SC) fraction of the sample (see text). The expectations for both Born and unitarity theories are also shown as dotted curves.

known to be negligible in $(TMTSF)_2ClO_4$ from previous $C_e/T$ or $H_{c2}(T)$ measurements[14,16,17]. This is also supported by the simple electronic structure consisting of essentially one band[41,42]. The magnetic impurity contribution may become important in magnetic fields, but we use only zero-field data in the present study. The nuclear specific heat of $(TMTSF)_2ClO_4$ in the present temperature range is known to be negligibly small at zero field. Thus, a simple extrapolation should be reliable for $(TMTSF)_2ClO_4$.

Another way to characterize the strength of the scattering process, free from any possible ambiguities originating from the extrapolation procedure, is through the $T_c$ dependence of the specific heat jump at the transition[43]. We estimated this jump by linearly extrapolating the low temperature $C_e/T$ up to $T_c$. The corresponding data points are shown with open symbols in Fig. 3(b).

## IV. DISCUSSION

### A. Born-like limit in $(TMTSF)_2ClO_4$

Figure 3(a) reveals a remarkable behaviour for the residual DOS. First, we note that $\gamma_0$ does not increase as $T_c$ decreases at larger cooling rates. This behaviour is at variance with most experiments where the unitarity limit is observed and induces a fast increase of $\gamma_0$ at increasing scattering[32,44–46]. This is a first sign that the random $ClO_4$ anions in $(TMTSF)_2ClO_4$ behave as Born-limit scatterers.

However, as shown in Fig. 2(b), for cooling rates larger than about 1 K/min, we have to take into account that only a fraction of the sample undergoes the superconducting transition due to the formation of granular superconductivity[22]. As a consequence, the measured $\gamma_0$ results from the contributions of both the superconducting and the normal parts of the sample:

$$\gamma_0 = \gamma_{normal} + \gamma_{super} \tag{1}$$
$$= (1-p)\gamma_N + p\gamma_{0,s}, \tag{2}$$

where $\gamma_{0,s}$ is the residual quasiparticle DOS for the superconducting part of the sample. At this point, let us emphasize that, at the cooling rates considered here, the SDW phase is not stablized, and its contribution to the low temperature specific heat can be neglected.[22,39,47] The volume fraction $p$ that intervenes in Eq. (2) is not straightforward to determine. A minimum value for $p$ is the volume fraction of the ordered regions $p_{ordered}$ as determined by the normal state resistivity behavior[22]. A maximum value for $p$ is the total superconducting volume fraction $p_{shielded}$ as determined from the fit to the data of Fig. 2(b)[22]. Both limits are shown in Fig. 4(a).

The current understanding of granular superconductivity in $(TMTSF)_2ClO_4$ is that, at cooling rates above 1 K/min, ordered regions become superconducting and couple by proximity effect. This induces superconducting coherence in regions that would otherwise remain normal. The difference between $p_{shielded}$ and $p_{ordered}$ corresponds to these disordered regions which carry supercurrents and are thus shielded. However, determining the contribution of these regions to the low temperature DOS is difficult.

Indeed, in the proximity effect with $s$-wave superconductors, proximitized regions exhibit a superconducting gap within the normal-state coherence length[48]. For $d$-wave superconductors, however, the density of states in the proximitized region depends on the orientation of the normal-superconductor interface relative to the superconducting order parameter and may develop an $s$-wave component near the interface[49–53]. In the following, we will therefore consider that $p_{ordered} \leq p \leq p_{shielded}$.

Let us compare our data with theoretical predictions more quantitatively. For example, at $T_c/T_{c0} = 0.65$ (cooling rate = 4.7 K/min), we have $\gamma_0/\gamma_N \simeq \gamma_{0,s}/\gamma_N \simeq 0.05 \pm 0.1$. This value is compatible with the Born limit, and clearly incompatible with the unitarity limit,

for which $\gamma_{0,s} \simeq 0.6\gamma_N$ is expected. We comment here that, up to this cooling rate, the correction due to volume fraction change is negligible since the experimentally obtained $\gamma_0$ is close to zero. For the fastest cooling rate (16.8 K/min, $T_c/T_{c0} = 0.48$), we need to take into account the change in the volume fraction. The actual $p$ should be in the range between $p_{ordered} \simeq 0.4$ and $p_{shielded} \simeq 0.55$. In the unitarity limit theory, we would expect $\gamma_{0,s,U} \simeq 0.75\gamma_N$. After considering the effect of the decrease in the volume fraction using Eq. (2), this value of $\gamma_{0,s,U}$ leads to $\gamma_0/\gamma_N \in [0.85, 0.90]$ . In the Born limit, we would expect $\gamma_{0,s,B} \simeq 0.2\gamma_N$, leading to $\gamma_0/\gamma_N \in [0.55, 0.70]$. Thus, the experimental value of $\gamma_0/\gamma_N \simeq 0.45 \pm 0.1$ is more compatible with the Born limit. Whether the scattering is truly in the Born limit or merely Born-like remains to be determined by future work, but we believe that the data presented in this work clearly show that the unitarity limit is not realized at intermediate disorder. Instead, the effective scattering character becomes Born-like as disorder increases, indicating a crossover in dominant scattering processes.

As another way to characterize the scattering process, we focus on the $T_c$ dependence of the specific-heat jump at the transition. As calculated first by Suzumura and Schulz, and then by Puchkaryov and Maki[27,30], the specific-heat jump $\Delta C$ at the transition normalized by $\gamma_N T_c$ is larger in the Born limit than in the unitarity limit. The experimental data (Fig. 3(b)) after renormalized by $p$ to account for the change in the superconducting fraction, is close to the Born limit. This data on the specific-heat jump provides an additional confirmation for the Born-limit behavior. Let us moreover stress that this analysis does not depend on the electronic specific heat extrapolation method at low temperatures.

As can be seen, both the residual density of states $\gamma_0$ and the specific heat jump at the superconducting transition $\Delta C$ suggests that the Born limit more adequately describes the effect of scattering centers on the superconductivity in $(\text{TMTSF})_2\text{ClO}_4$ at large disorder.

## B. Evolution of pair breaking mechanism with disorder

In this subsection, we discuss an additional surprising feature in Fig. 3(a). Up to about 6 K/min, that is until $T_c$ drops to about $0.65\,T_{c0}$, the effect of the volume-fraction change is negligible. Under the slowest cooling rate of this experiment (30 mK/min, $T_c/T_{c0} = 0.85$), $\gamma_0/\gamma_N \approx 0.2$. This value is consistent with the NMR data[15,40] (green point) measured under cooling rate of 7 mK/min. Then, $\gamma_0$ shows a slight tendency to decrease as $T_c$ decreases. Finally, $\gamma_0$ experiences an upturn above 10 K/min, mainly due to the normal contribution becoming dominant ($p \approx 50\%$). Interestingly, a weak minimum in $\gamma_0/\gamma_N$ exists between both trends. This minimum points to the existence of two different regimes for impurity scattering. We propose below a tentative inter-

pretation for this feature.

First, preexisting chemical impurities, in particular in the TMTSF conduction layer, are unavoidable in the chemical synthesis of such materials. At our slowest possible cooling rate (30 mK/min), $T_c/T_{c0} = 0.85$ can be related to a residual DOS of $\gamma_0 \approx 0.2\gamma_N$. This is likely due to such chemical impurities. These can be magnetic and could therefore act as strong dilute scatterers[54–56]. Indeed, the non-zero $\gamma_0$ at slow cooling rates can be interpreted as the sign that this scattering by chemical impurities is in the unitarity limit. We presume that this impurity-induced state by chemical impurities is located close to zero energy and extended along the SC gap node direction in real space, as illustrated in Ref.[57] for d-wave superconductivity.

We here comment on the possibility of mesoscopic phase segregation between normal and superconducting parts within a sample even under the slowest cooling, thereby contributing to an apparent residual DOS. It is difficult to rule out this possibility completely. However, the presence of a small non-superconducting fraction would simply introduce an additive offset to the extracted $\gamma/\gamma_0$ values. It would not change the overall trend of Fig. 3(a). Thus the main conclusion of this paper is not altered.

As the cooling rate is increased, $\gamma_0/\gamma_N$ decreases with $T_c/T_{c0}$. This is surprising since the cooling rate is not expected to affect the concentration of pre-existing localized impurities. In a naive picture, we would expect $\gamma_0/\gamma_N$ to be constant.

Although a full theory of the effect of extended defects remains, to the best of our knowledge to be developed, our interpretation of this phenomenon is that a crossover in the nature of the pair-breaking process takes place as soon as the cooling rate departs from the slowest one. High-resolution X-ray investigations[58–60] have shown that, with increasing cooling rates, $(\text{TMTSF})_2\text{ClO}_4$ organizes itself into finite-sized anion-ordered domains, inserted into non-superconducting anion-disordered background. As a consequence, pair-breaking could be expected at the non-magnetic borders of ordered domains. This mechanism is known to be sensitive to cooling rate[58]. Note that X-ray studies have shown that lone misaligned $\text{ClO}_4$ anions are scarce, so that scattering against those is unlikely to be the dominant process. We therefore propose that scattering becomes progressively governed by that on the boundaries of anion-ordered regions. The steady decrease of the residual DOS when $T_c/T_{c0}$ changes from 1 to 0.6 leads us to propose that this scattering mechanism is weak and follows the Born limit.

To support this, from Fig. 2(c), we have extracted the normalized electronic mean free path $\lambda/\lambda_c$ from the scattering rate, assuming that $\Gamma/\Gamma_c = \lambda_c/\lambda$, with $\lambda_c$ the mean free path at which $T_c = 0$ (Fig. 4(c)). It is remarkable that it follows a cooling-rate dependence very similar to the size of the anion-ordered domains $L_{2D}$ in the $ab$ plane observed by X-ray scattering experiments[58].

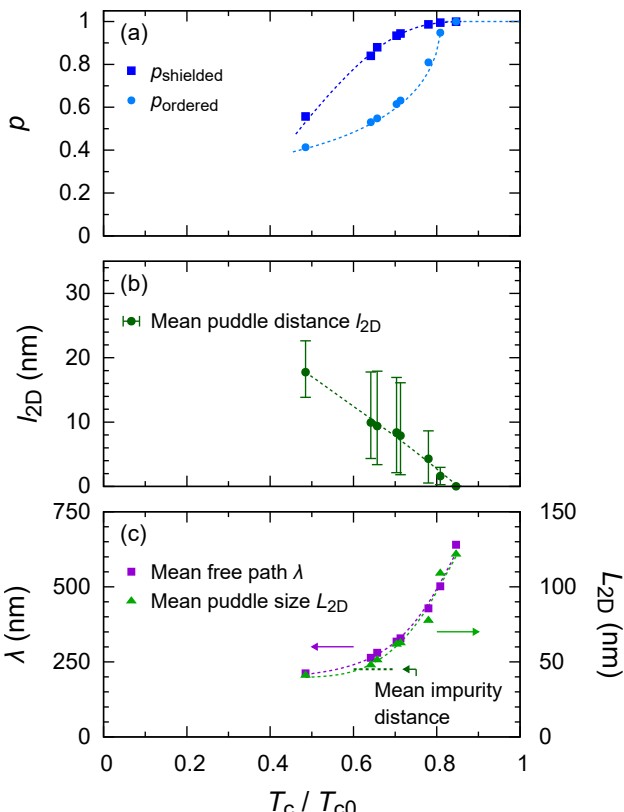

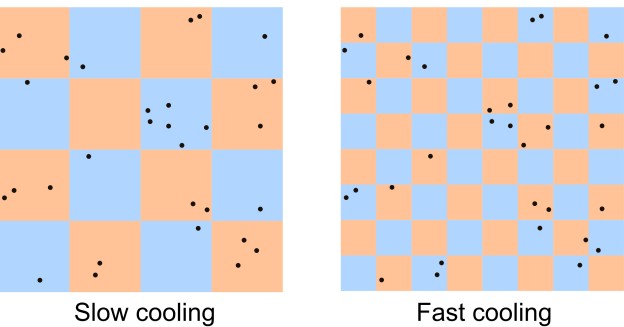

FIG. 5: Crude 2D picture of anion-ordered domains (blue and orange) in a background of randomly distributed localized chemical impurities, represented as black dots. The fraction of domains containing no impurity is very small under slow cooling conditions (left), but increases significantly at rapid cooling (right) as the size of domains decreases.

FIG. 4: (a) $T_c/T_{c0}$-dependence of $p_{shielded}$, as extracted from Fig. 2(b), and of $p_{ordered}$ as extracted from the normal state resistivity (Ref. 22). (b) Mean inter-puddle distance $l_{2D}$ extracted from Eq. (3). The bars correspond to the two extreme values for $p$: $p_{shielded}$ and $p_{ordered}$. (c) Mean ordered domain size $L_{2D}$ in the $ab$ plane[58] and mean free path $\lambda$ as extracted from the scattering rate (Fig. 2(c)). The horizontal dashed line shows the cooling rate at which the average domain size amounts to the distance between local defects for a concentration of 20 ppm/unit cell. All lines are guides to the eye.

At high cooling rates, the distance $l_{2D}$ between ordered domains increases fast. This can be understood in terms $L_{2D}$. Assuming, crudely, that the ordered domains are squares in the $ab$ plane, $l_{2D}$ in the $ab$ plane is given by

$$l_{2D} \simeq \frac{1 - \sqrt{p}}{\sqrt{p}} L_{2D}. \qquad (3)$$

According to this picture, the minimum for $\gamma_0$ ($T_c/T_{c0} \simeq 0.65$) in Fig. 3 (signaled by the horizontal dashed line in Fig. 4(c)) may be interpreted as resulting from the crossover between the two dominant scattering mechanisms when the mean distance between local strong scattering defects becomes of the order of $L_{2D}$. Using the value of $L_{2D} = 45$ nm as determined by X-ray measurements[58] for a value of $T_c/T_{c0} \simeq 0.62$, we can determine that the local defect concentration would be of $\approx 0.001\%$/unit cell. This value is in line with val-

ues reported in the literature, namely $< 0.008\%$/mole[58]. Furthermore, on the basis of the EPR linewidth dependence of e-beam irradiated $(TMTSF)_2ClO_4$ at very low temperature, the amount of residual spins per mole of TMTSF in a pristine sample is inferred to be at most $0.05\%$/mole[61].

How both scattering mechanisms act is illustrated in Fig. 5 using a cartoon picture of randomly-arranged localized chemical impurities superimposed on a chessboard lattice mimicking anion-ordered domains. The concentration of chemical impurities is constant for a given sample. However, the density of boundaries between ordered domains varies with the cooling rate. At slow cooling rates, the domain size are large, so that the concentration of domains which are free from impurities is low. All domains are therefore significantly affected by chemical impurities. Increasing the cooling rate makes the averaged domain size smaller, leading in turn to a larger fraction of ordered domains without any impurities. Those are likely to retain a $T_c$ unaffected by strong scatterers. For instance, if the distance between two local impurities is $\sim 40$ nm at $T_c/T_{c0} \simeq 0.65$ as discussed above, this would translate into $\sim 10$ local defects per ordered zone, on average, at the slowest cooling rate of 30 mK/min. We believe that this accounts for the larger values of $\gamma_0$ at slow cooling rates.

Theories predict the suppression of $d$-wave pairing[25] for a critical mean free path $\lambda_c = \pi \xi_{ab}$, where $\xi_{ab} = 45$ nm is the superconducting coherence length in the $ab$ plane[62]. Assuming that $\lambda_c$ corresponds to $\Gamma_c$ and according to Fig. 2(c), one obtains mean free paths of about 210 nm and 650 nm at 16.8 K/min and 30 mK/min, respectively (Fig. 4(c)). This is about 5 times larger than the typical size of ordered domains. This means that the scattering process is due to passing through domain walls, and a pair has to travel through typically 5 domain walls to loose its coherence, consistent with the Born limit picture developed here.

## V. CONCLUSION

Although Born-limit scattering was already discussed by theoreticians in the context of cuprates in the early 1990s[26,29,63], direct experimental evidence has remained scarce. Over time, however, it has become clear that the effective scattering potential in unconventional superconductors is usually intermediate between the Born and unitary limits[33]. The true endpoints of purely Born or purely unitary scattering are likely never realized in practice. In this context, we believe that $(TMTSF)_2ClO_4$, and more generally organic superconductors, provide a particularly interesting platform to experimentally investigate and test the applicability of these scattering models. Indeed, through this work, we investigated the effect of nonmagnetic impurities on the thermodynamics properties of the Q1D organic superconductor $(TMTSF)_2ClO_4$. We have shown that, at strong $ClO_4$ disorder, there is compelling evidence for the existence of Born-like scattering in this nodal $d$-wave-like superconducting system. At very low $ClO_4$ disorder, in contrast, we have shown that scattering may very well be dominated by isolated chemical impurities, possibly in the unitarity limit. Increasing non magnetic disorder there is evidence for a cross over towards a regime where scattering becomes dominated by a Born-like scattering controlled by the size of the anion-ordered domains, which, in turn, determines the elastic electron mean free path.

This proposed crossover would call for further investigation toward direct confirmation. For example, previous experimental investigations of the solid solution $(TMTSF)_2(ClO_4)_{(1-x)}(ReO_4)_x$ have shown that even a small concentration of $ReO_4$ impurities suppresses $T_c$ very efficiently[38]. Moreover, diffuse X-ray scattering[64] revealed that long-range $ClO_4$ order persists up to about $x = 3\%$. Since it is very likely that $ReO_4$ anions are surrounded by disordered domains acting as weak scattering centers for superconductivity, very much like the disordered domain walls in pure $(TMTSF)_2ClO_4$, the investigation of the superconducting thermodynamic properties of $(TMTSF)_2ClO_4$-$ReO_4$ alloys should be very valuable to derive the strength of the scattering induced by such an isoelectronic anion disorder.

## VI. ACKNOWLEDGMENTS

We gratefully acknowledge stimulating discussions with Claude Bourbonnais, Christophe Brun, and Vincent Humbert. This project was partly funded by France-Japan bilateral joint funding: the PHC Sakura 2022 (project n°48320WB) for the French side and Grant-in-Aid for Bilateral Joint Research Projects (No.JPJSBP120223205) from the Japan Society for the Promotion of Science (JSPS) for the Japanese side. The work at Kyoto Univ. was additionally supported by Grant-in-Aids for Scientific Research on Innovative Areas "Quantum Liquid Crystals" (KAKENHI Grant Nos. 20H05158, 22H04473a) from JSPS, Grant-in-Aids for Academic Transformation Area Research (A) "1000 Tesla Science" (KAKENHI Grant No. 23H04861) from JSPS, by a research support funding from The Kyoto University Foundation, by ISHIZUE 2020 and 2023 of Kyoto University Research Development Program, and by The Murata Science Foundation. S. Yonezawa acknowledges support for the construction of the calorimeter from Research Equipment Development Support Room of the Graduate School of Science, Kyoto University; and support for liquid helium supply from Low Temperature and Materials Sciences Division, Agency for Health, Safety and Environment, Kyoto University.

## Appendix A: Calorimeter design

To measure the specific heat, we have built a custom-made calorimeter that is compatible with the PPMS-ADR cryostat. A bare chip of a resistive thermometer (Lakeshore, Cernox) was used as the sample stage. The thermometer element was cut into two parts by focused ion beam (FIB). One part is used as a thermometer and the other as a heater. This construction minimizes the background contribution. We used Pt-W wires with a diameter of 25 $\mu$m to construct four-wire connections to the thermometer and heater, as well as to hang the sample stage. Silver epoxy (Epotek H20E) was used for electrical connection. The thermometer was calibrated before the heat-capacity measurement by thermally contacting the sample stage and the thermal bath with a gold foil.

## Appendix B: Cool-down procedure

To control the anion ordering, we first cool down the system to 50 K and keep the temperature constant for one hour to fully randomize the $ClO_4$ anion orientation. Subsequently, the system was cooled down with a fixed cooling rate down to 10 K. The final step was to cool the sample down to the lowest temperature using the adiabatic demagnetization procedure. For the 16.8 K/min data, the system was cooled first to 10 K under high vacuum to keep the sample temperature to 50 K, before putting $\sim$ 200 Pa of helium exchange gas to start rapid cooling. All cooling rates were calculated by a linear fitting of the time dependence of the sample-stage temperature around $T_{AO}$.

## Appendix C: AC specific heat measurement

We used the AC calorimetry method to measure the small heat capacity of the sample[36]. To determine which frequency $\omega_H$ to use for the current injected in the

heater, we measured the temperature dependence of the heat capacity under various frequencies and chose a frequency maximizing the sensitivity while avoiding extrinsic frequency-dependent results. In these experiments, we used $\omega_H/2\pi = 21.14$ Hz. The amplitude of the heater current is chosen so that the temperature oscillation amplitude is about 1% of the sample temperature.

We also performed measurements of the background heat capacity of the sample stage. We found that determining the proper $\omega_H$ for the background (addenda) measurement was complicated by a spurious frequency dependence at higher frequency. Thus, we fitted the low-frequency data below 32 Hz to obtain the heat capacity $C$ by using the theoretical formula[36]

$$\frac{T_{AC}}{P_0} = \frac{1}{2\omega_H C} \frac{1}{\sqrt{1 + (\tau_1 \omega_H)^{-2} + (\tau_2 \omega_H)^2}}.$$

From the raw heat capacity data, after subtracting this background contribution, the phonon contribution was obtained by fitting $C/T = \gamma + C_{\text{phonon}}/T$ to the data above $T_c$, where $C_{\text{photon}}/T$ is assumed to have the standard form $\beta T^2$. Although this phonon model is very simple, we found that the model with one fixed value of $\beta$ fits the data well throughout all the cooling rates. The obtained value of $\beta = 11.86$ mJ/K$^4$ mol correspond to the Debye temperature $\Theta_D$ of 211 K, which is consistent with a previous result ($\Theta_D = 213$ K; Ref. 65). For $\gamma$, we allow this value to vary for different cooling rates, but the resultant change in $\gamma$ is less than 0.5 mJ/mol K$^2$ ($\sim 3\%$ of the total $\gamma$ value). Also, we should comment that in-field specific heat measurement to estimate the normal-state contribution is not straightforward when using an ADR, where the condition $H \simeq 0$ is necessary for maximum cooling down. Thus, we judge that use of this model for the normal-state contribution is adequate for the present study. Even after subtracting $C_{\text{phonon}}/T$, the data was found to contain a small peak at around 0.8 K. This peak was independent of the cooling rate and a similar peak is seen in the background contribution. Thus, we judged that this small peak is extrinsic, probably originating from under-substraction of the background. For the slowest-cooled data, this peak was fitted with two gaussian functions plus a polynomial function representing the intrinsic heat capacity, and these gaussian functions are subtracted from all the data.

### Appendix D: Extrapolation of $C_e/T$ in the $T \to 0$ limit

To extrapolate the residual electronic specific heat for $T \to 0$, we need to assume a theoretical function. However, since different theoretical models predict different temperature dependence for $C_e/T$ at low temperature[27,30,66], there is an ambiguity in the choice of the extrapolation function. We have therefore tried to use extrapolation functions $f_\alpha(T) = a_0 + a_1 T + a_2 T^\alpha$ by systematically changing the exponent $\alpha$ from 1.1 to 3.0,

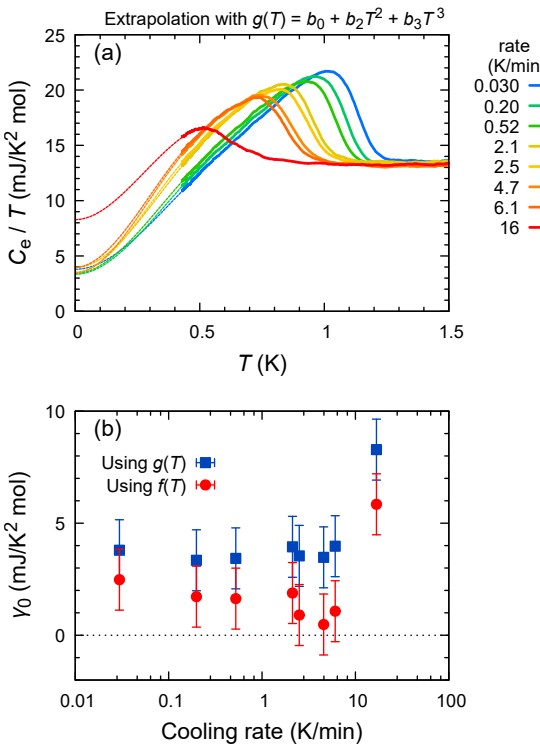

FIG. 6: Test with another extrapolation model. (a) Extrapolation of $C_e/T$ data using $g(T) = b_0 + b_2 T^2 + b_3 T^3$. The solid curves are experimental data and the broken curves show extrapolation. (b) Comparison of cooling-rate dependence of the extrapolated $\gamma_0$ values obtained with different functions. Although there is a constant offset between extrapolations using $g(T)$ (blue squares) and $f(T) = a_0 + a_1 T + a_2 T^2$ (red circles), the overall trend is similar in both datasets.

although $\alpha = 2$ seems to be the most theoretically valid value since this exponent corresponds to the theoretical model of $C_e/T$ of line-nodal superconductors with Born impurities[27]. We have also tried other extrapolation functions such as $g(T) = b_0 + b_2 T^2 + b_3 T^3$, which is valid for line-nodal superconductors with unitarity impurity scattering[67]. As an example, results of the extrapolation using $g(T)$ are shown in Fig. 6. As shown in (a), this extrapolation works well. In the panel (b), we compare the extrapolated $\gamma_0$ values obtained from extrapolations using the two different model functions $f(T)$ and $g(T)$. The overall trend of $\gamma_0$ as a function of the cooling rate remains the same, besides a constant offset between the two datasets. We have performed similar examinations using other model functions mentioned above, and confirmed that the trend remains irrespective of the extrapolation model. Thus the overall qualitative dependence of the residual DOS on $T_c$ remains the same as the one described in Fig. 3, and therefore the main conclusion of this paper on the residual DOS is independent of the extrapolation uncertainties.

* Electronic address: claire.marrache@universite-paris-saclay.fr

† Electronic address: denis.jerome@universite-paris-saclay.fr

‡ Electronic address: yonezawa.shingo.3m@kyoto-u.ac.jp

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
