# Peer review of "Born-limit scattering and pair-breaking crossover in d-wave superconductivity of (TMTSF)2ClO4"

_SciPost Physics_

## Round 1 · Referee Report · Anonymous (Referee 1) · 2025-6-30

Strengths

The manuscript presents a study of the effects of nonmagnetic disorder on the specific heat of unconventional superconductor (TMTSF)₂ClO₄. The authors conclude that the only way to distinguish between Born and unitary scattering limits is to examine the residual density of states, obtained from the specific heat.

Weaknesses

There are a number of issues.
1. The manuscript states: “However, while this extension predicts the effect of increased scattering on Tc , it does not provide any information how scattering acts on the superconducting order parameter.” This statement is incorrect. For any superconductor, conventional or unconventional, Tc is always proportional to the magnitude of the order parameter, just with different pre-factors.
2. The discussion of the density of states, N(E,k) ( DOS) in the introduction should be more specific. Heat capacity includes the integral over the Fermi surface containing N(E,k) of DOS, so it includes all energies. Unitary scattering primarily affects the states at energies close to the Fermi level, whereas Born limit influences the states near the gap edge, see V. G. Kogan, R. Prozorov, and V. Mishra, London penetration depth and pair breaking, Phys. Rev. B 88, 224508 (2013). Therefore DOS is affected quite significantly in both regimes. It is true, however, that in the Born limit, a very significant scattering rate is required to suppress Tc and superfluid density, hence affecting any thermodynamic quantity, such as heat capacity. This is the method the manuscript uses to distinguish between two regimes.
3. The authors state that “Almost all unconventional superconductors follow…” unitary limit. This may not quite accurate. Historically, this was the sole theory explaining a rapid change of Tc with non-magnetic disorder in high-Tc. Somehow the Authors do not cite this seminal paper, P. J. Hirschfeld and N. Goldenfeld, Effect of strong scattering on the low-temperature penetration depth of a d-wave superconductor, Phys. Rev. B 48, 4219 (1993).
Over time, it has been recognized that the scattering potential strength is usually intermediate, see for example (and references therein), K. Cho, M. Kończykowski, S. Teknowijoyo, S. Ghimire, M. A. Tanatar, V. Mishra, and R. Prozorov, Intermediate scattering potential strength in electron-irradiated YBa$2$Cu$3$O$ _{7-\delta}$ from London penetration depth measurements, Phys. Rev. B 105, 14514 (2022). This generalized potential is treated withing the well-developed T-matrix approach. It is believed that end-points of the scattering potential (Born and unitary limits) are never realized and the scattering strength is somewhere in between.
4. For a generalized theory of the effects of arbitrary scattering and order parameters, I recommend consulting L. A. Openov's work: Effect of nonmagnetic and magnetic impurities on the specific heat jump in anisotropic superconductors, Phys. Rev. B 69, 224516 (2004).
5. Stylistically, it appears the authors use “resonant scattering” and “unitary scattering” interchangeably, which are distinct regimes. The former produces in-gap bound states, while the latter affects DOS smoothly, as explained in the first referenced paper above.
6. The authors apply an over-simplified background subtraction developed for isotropic metals with isotropic pairing. Phonon modes are sensitive to any anisotropy. More critically, the authors assume there is no electronic contribution above Tc, stating: “after subtracting this background contribution, the phonon contribution was obtained by fitting Cphonon/T = B22T^2+B44*T^4 to the data above Tc.” It is likely they used, but neglected to include, the electronic term (which would be a constant) in the text. This procedure has significant uncertainty due to the fitting region being far from T=0, and because the formula is derived for a simple Fermi-liquid model, which does not apply to many unconventional superconductors.
Usually, phonons are subtracted by suppressing superconductivity using a magnetic field, though magnetoresistance is a problem. (Another way is to use a non-magnetic analog of the studied compound, but this is inapplicable here.) Given the reliance on the subtraction procedure, I do not know how reliable the electronic part is. This important point should be addressed in detail.
7. Another concern is that the manuscript shows only normalized scattering rates, which is obtained from the normalized resistivity Ref.[33], see Fig.2. This is misleading. The authors should derive the absolute values of the scattering rate, \Gamma, and plot it along with the AG-Openov curve for a d-wave superconductor with non-magnetic impurities, which has a specific critical value of \Gamma_c=0.28 when Tc is suppressed to zero. Depending on gap anisotropy, the function Tc(\Gamma) can be anywhere from a scattering-independent constant to the AGO curve.
8. All mentioned theories assume point-like scattering potential. The model suggested in the manuscript involves domain boundaries. Scattering on extended defects is considerably more complex has very different effect on properties .

Report

It is evident that considerable effort was put into this work, and the authors did their best to present and analyze the data. However, the significant issues outlined above need to be addressed before considering it for any journal.

Requested changes

The changes must be substantial, impossible to list here.

Recommendation

Reject

  • validity: low
  • significance: good
  • originality: ok
  • clarity: ok
  • formatting: good
  • grammar: good

Author:  Claire Marrache  on 2025-09-15  [id 5819]

(in reply to Report 1 on 2025-06-30)

We would like to thank the reviewers for their careful review of our manuscript. Below are our answers to their concerns and the summary of the changes made in the manuscript to address them.

Report #1 Strengths "The manuscript presents a study of the effects of nonmagnetic disorder on the specific heat of unconventional superconductor (TMTSF)₂ClO₄. The authors conclude that the only way to distinguish between Born and unitary scattering limits is to examine the residual density of states, obtained from the specific heat." We thank Reviewer #1 for approving the method at the core of our paper.

Weaknesses There are a number of issues. 1. "The manuscript states: “However, while this extension predicts the effect of increased scattering on Tc , it does not provide any information how scattering acts on the superconducting order parameter.” This statement is incorrect. For any superconductor, conventional or unconventional, Tc is always proportional to the magnitude of the order parameter, just with different pre-factors. " We admit that the latter half of this sentence was misleading. We meant that measuring Tc as a function of scattering does not provide any information on whether the scattering is weak or strong. Thus, we have modified this sentence in the revised manuscript so that it is clearer. The text now reads “However, while this extension predicts the effect of increased scattering on Tc, it does not provide any information on whether the scattering is weak or strong”.

  1. "The discussion of the density of states, N(E,k) ( DOS) in the introduction should be more specific. Heat capacity includes the integral over the Fermi surface containing N(E,k) of DOS, so it includes all energies. Unitary scattering primarily affects the states at energies close to the Fermi level, whereas Born limit influences the states near the gap edge, see V. G. Kogan, R. Prozorov, and V. Mishra, London penetration depth and pair breaking, Phys. Rev. B 88, 224508 (2013). Therefore DOS is affected quite significantly in both regimes. It is true, however, that in the Born limit, a very significant scattering rate is required to suppress Tc and superfluid density, hence affecting any thermodynamic quantity, such as heat capacity. This is the method the manuscript uses to distinguish between two regimes." We thank Reviewer #1 for the helpful clarification regarding the energy dependence of the density of states N(E, k) and its role in modifying the heat capacity. As Reviewer #1 notes, heat capacity integrates in the energy range of the order of kBT and is therefore a relevant quantity for capturing the effects of disorder and distinguish between unitary and Born scattering regimes. We appreciate Reviewer #1’s recognition that, in the Born limit, a large scattering rate is needed to noticeably impact thermodynamic quantities such as heat capacity — a point that underlies our method for distinguishing between the two regimes. We have added the reference provided by the reviewer and thank him/her for bringing this paper to our attention. In the revised manuscript, we have clarified this point in the introduction to better convey why heat capacity is an appropriate and sensitive probe in this context.

  2. "The authors state that “Almost all unconventional superconductors follow…” unitary limit. This may not quite accurate. Historically, this was the sole theory explaining a rapid change of Tc with non-magnetic disorder in high-Tc. Somehow the Authors do not cite this seminal paper, P. J. Hirschfeld and N. Goldenfeld, Effect of strong scattering on the low-temperature penetration depth of a d-wave superconductor, Phys. Rev. B 48, 4219 (1993). Over time, it has been recognized that the scattering potential strength is usually intermediate, see for example (and references therein), K. Cho, M. Kończykowski, S. Teknowijoyo, S. Ghimire, M. A. Tanatar, V. Mishra, and R. Prozorov, Intermediate scattering potential strength in electron-irradiated YBa2Cu3O7−δ from London penetration depth measurements, Phys. Rev. B 105, 14514 (2022). This generalized potential is treated within the well-developed T-matrix approach. It is believed that end-points of the scattering potential (Born and unitary limits) are never realized and the scattering strength is somewhere in between." We thank Reviewer #1 for pointing out this important historical context and for highlighting the relevant references. We have added the seminal work by Hirschfeld and Goldenfeld [Phys. Rev. B 48, 4219 (1993)] as well as the more recent study by Cho et al. [Phys. Rev. B 105, 014514 (2022)] to the revised manuscript. Accordingly, we have modified our original statement to acknowledge that some unconventional superconductors exhibit scattering strengths that are intermediate between the Born and unitary limits, rather than strictly adhering to the unitary regime. We would like to emphasize that much of the insight into scattering regimes has come from London penetration depth studies, whereas our own measurements are thermodynamic (heat capacity). We now emphasize that the Born and unitary limits are useful idealized cases and serve as endpoints of a continuum of scattering strengths. A complete theoretical description that captures the effective scattering potential in effectively granular materials (such as (TMTSF)2ClO4) or intermediate regimes remains, to the best of our knowledge, to be developed. We hope these changes improve the clarity and balance of the discussion.

  3. "For a generalized theory of the effects of arbitrary scattering and order parameters, I recommend consulting L. A. Openov's work: Effect of nonmagnetic and magnetic impurities on the specific heat jump in anisotropic superconductors, Phys. Rev. B 69, 224516 (2004). " We thank Reviewer #1 for bringing the work of L. A. Openov [Phys. Rev. B 69, 224516 (2004)] to our attention. We were not previously aware of this reference and have now added it to the manuscript. Openov’s analysis of the specific heat jump in the presence of nonmagnetic and magnetic impurities, particularly in mixed s- and d-wave order parameters, provides valuable context. As far as we understand, this work does not address the residual density of states directly, and provides predictions for the specific heat jump. Our experimental data are consistent with a purely d-wave order parameter, corresponding to the case r=0 in Openov’s model.

  4. "Stylistically, it appears the authors use “resonant scattering” and “unitary scattering” interchangeably, which are distinct regimes. The former produces in-gap bound states, while the latter affects DOS smoothly, as explained in the first referenced paper above." We thank Reviewer #1 for pointing out this. We had indeed used “resonant scattering” once in the conclusion. We’ve modified it to “unitarity scattering” to align with Maki et al.’s denomination.

  5. "The authors apply an over-simplified background subtraction developed for isotropic metals with isotropic pairing. Phonon modes are sensitive to any anisotropy. More critically, the authors assume there is no electronic contribution above Tc, stating: “after subtracting this background contribution, the phonon contribution was obtained by fitting Cphonon/T = B22T^2+B44T^4 to the data above Tc.” It is likely they used, but neglected to include, the electronic term (which would be a constant) in the text. This procedure has significant uncertainty due to the fitting region being far from T=0, and because the formula is derived for a simple Fermi-liquid model, which does not apply to many unconventional superconductors. Usually, phonons are subtracted by suppressing superconductivity using a magnetic field, though magnetoresistance is a problem. (Another way is to use a non-magnetic analog of the studied compound, but this is inapplicable here.) Given the reliance on the subtraction procedure, I do not know how reliable the electronic part is. This important point should be addressed in detail." First of all, we apology the mistype in the previous manuscript. Actually, we did include the electronic term in the fitting as a standard procedure. The part mentioned by Reviewer #1 should have been written as “after subtracting this background contribution, the phonon contribution was obtained by fitting C/T = gamma + Cphonon/T to the data above Tc, where Cphonon/T is assumed to have the form B22T^2 + B44T^4”. We now modified the 2nd paragraph of Appendix C. I hope this clarifies most of Reviewer’s concern. It is true that suppressing superconductivity by magnetic field is often used to estimate phononic contribution. Nevertheless, it is not straightforward to perform in-field measurements in the present setup using an adiabatic demagnetization refrigerator, where zero-field condition is necessary to cool down the sample. We add a short explanation on this point in the 2nd paragraph of Appendix C. On the influence of the anisotropy of phonon modes, although we agree that our model is a very simple one, but we are not aware of any widely-accepted and reliable model to include such effects to analyze the specific-heat data. We would like to emphasize that the simple model for the phonon contribution fits the normal-state data well irrespective of the cooling rate. We also comment that more complicated models may have risks of overfitting of the data. We thus believe that the simple model is phenomenologically good enough for our purpose. To address Reviewer #1’s concern, we added a short description in the 2nd paragraph of Appendix C.

  6. "Another concern is that the manuscript shows only normalized scattering rates, which is obtained from the normalized resistivity Ref.[33], see Fig.2. This is misleading. The authors should derive the absolute values of the scattering rate, \Gamma, and plot it along with the AG-Openov curve for a d-wave superconductor with non-magnetic impurities, which has a specific critical value of \Gamma_c=0.28 when Tc is suppressed to zero. Depending on gap anisotropy, the function Tc(\Gamma) can be anywhere from a scattering-independent constant to the AGO curve." It is true that evaluation of the absolute value of Γc is ideal, but experimentally, accurate quantitative evaluation of Γc is often very difficult. Instead, in many experimental papers of various unconventional superconductors, the ratio Γ/Γc (or just the residual resistivity) is used to compare experimental data with the AG framework. The difficulty in finding Γc is exemplified by the fact that Γc is not universal and depends on the nature of impurities. In the (TMTSF)2ClO4 system, this is corroborated by the following references: Joo et al. [Eur. Phys. J. B 40, 43 (2004)] reports the critical residual resistivity of ρc ≈ 0.185 Ω⋅cm for (TMTSF)2ClO4/ReO4 solid solutions, while for pure (TMTSF)2ClO4, Yonezawa et al. [Phys Rev B 97, 014521 (2018)] reports ρc ≈ 0.24 Ω⋅cm. For the present study, our fitting in Fig.2(c) yields 0.14 Ω⋅cm (this small value may be attributable to the fact that bulk thermodynamic quantity is used in this study, whereas resistivity is used to evaluate Tc in the previous two studies). Thus, Γc, which is proportional to ρc, would differ by 170% among different data sets. Because of this fact, as well as many examples of previous papers on unconventional superconductors, the use of Γ/Γc in the present experimental paper should be justified. To address Reviewer’s concern, we added the value of the critical resistivity in the last paragraph of Sec. IIIA. We emphasize that, besides the difference in the critical scattering, Tc dependence on the scattering is quite well explained by the AG theory for d-wave superconductivity influenced by non-magnetic impurities.

  7. "All mentioned theories assume point-like scattering potential. The model suggested in the manuscript involves domain boundaries. Scattering on extended defects is considerably more complex has very different effect on properties." We thank Reviewer #1 for this important observation. Indeed, most standard theoretical treatments, including those we reference, assume point-like impurity scattering. In contrast, the model we consider involves scattering from extended defects such as domain boundaries, which are inherently more complex and can influence quasiparticle properties differently. We fully agree that a comprehensive theory accounting for extended scatterers is still lacking and would be necessary to quantitatively capture all effects. Our intention was to provide a phenomenological framework motivated by the experimental trends, while acknowledging that the microscopic scattering mechanisms—particularly in the granular regime—are likely to differ from those of point-like impurities.

Report "It is evident that considerable effort was put into this work, and the authors did their best to present and analyze the data. However, the significant issues outlined above need to be addressed before considering it for any journal." We sincerely appreciate Reviewer #1’s recognition of the effort invested in this experimental work. We would like to emphasize that our primary focus is on presenting and analyzing new experimental data, with the aim of stimulating further theoretical developments. We also mention that most serious concerns of Reviewer #1 are due to our writing issue and are now fully resolved. Thus, we now believe that the manuscript is ready for consideration of publication.

---

## Round 1 · Referee Report · Anonymous (Referee 2) · 2025-7-7

Strengths

The system under study lends itself to study the effect of (variable density, in situ) disorder on the residual density of states simultaneously with the effect on the superconducting transition temperature.

Weaknesses

  1. The argument given for explaining the drop in apparent DOS on defect density is unclear, and Fig. 3 only marginally helps in the explanation.
  2. The starting point of "strong scatterers" is not clearly defined (see the Report below).

Report

The manuscript entitled “Born-limit scattering and pair-breaking crossover in d-wave superconductivity of (TMTSF)2ClO4” (Yano, et al.) documents simultaneously the rise of the residual DOS and the depression of the SC critical temperature Tc with increased disorder. The residual DOS and Tc are determined from ac calorimetry measurements. The disorder is indirectly controlled by the rate of cooling through an anion ordering transition that takes place at higher temperature. The authors explain the Fermi surface is indirectly affected as a result since there is a period doubling of the unit cell along the b-direction with slow cooling.
From the variation of the residual DOS with cooling rate, the authors conclude that the defects introduced by increasing the cooling rate are Born-limit scatterers, and consequently very different than defects studied previously in unconventional superconductors. The claim is a strong one that requires explicit defense. My understanding is that neither limit applies generally.
The dependence on the residual DOS decreases with initial cooling rate. This is the most striking result in the article. The authors’ explanation for it is not clearly articulated, at least to this reader. My understanding of the effect of dilute strong (resonant) scatterers (the initial state, slow cooling case) in a d-wave SC is to introduce localized states extending out from the defect location like 4 arms (due to the gap nodal structure).

Is this the right starting point? The authors don’t clearly state it.

Of course, increasing the number of resonant scatterers would only increase the ratio \gamma_0/\gamma_N (defined in the article as the residual DOS relative to normal state DOS). Here, the point is: what about adding anion-orderered domain walls with increasing density? In order for the ratio \gamma_0/\gamma_N to decrease, I must assume the number of localized states also decreases. The authors have discussed the geometry of their model by introducing the Born scatterers as extended domain walls—a configuration not previously addressed to my knowledge. If they made an argument as to why this reduces the number of localized states, then I missed it.

Requested changes

The authors need to address the weaknesses identified above.

Recommendation

Ask for major revision

  • validity: -
  • significance: good
  • originality: good
  • clarity: poor
  • formatting: reasonable
  • grammar: good

Author:  Claire Marrache  on 2025-09-15  [id 5820]

(in reply to Report 2 on 2025-07-07)
Category:
answer to question
reply to objection

We would like to thank the reviewers for their careful review of our manuscript. Below are our answers to their concerns and the summary of the changes made in the manuscript to address them.

Report #2
Strengths
"The system under study lends itself to study the effect of (variable density, in situ) disorder on the residual density of states simultaneously with the effect on the superconducting transition temperature."
We thank Reviewer #2 for recognizing the value of (TMTSF)2ClO4, which disorder can be fine-tuned and thus serve as a model system for studying the scattering in d-wave superconductors.

Weaknesses
1. "The argument given for explaining the drop in apparent DOS on defect density is unclear, and Fig. 3 only marginally helps in the explanation. "
We assume Reviewer #2 refers to Fig.5? The main idea we are trying to convey is that, as the cooling rate changes, the density of chemical impurities is fixed, but the density of non-magnetic borders of ordered domains increases. This is what we have tried to illustrate in Fig 5. In the revised version of the manuscript, we have tried to be more explicit.

2. "The starting point of "strong scatterers" is not clearly defined (see the Report below)."
See response below.

Report
"The manuscript entitled “Born-limit scattering and pair-breaking crossover in d-wave superconductivity of (TMTSF)2ClO4” (Yano, et al.) documents simultaneously the rise of the residual DOS and the depression of the SC critical temperature Tc with increased disorder. The residual DOS and Tc are determined from ac calorimetry measurements. The disorder is indirectly controlled by the rate of cooling through an anion ordering transition that takes place at higher temperature. The authors explain the Fermi surface is indirectly affected as a result since there is a period doubling of the unit cell along the b-direction with slow cooling.
From the variation of the residual DOS with cooling rate, the authors conclude that the defects introduced by increasing the cooling rate are Born-limit scatterers, and consequently very different than defects studied previously in unconventional superconductors. The claim is a strong one that requires explicit defense. My understanding is that neither limit applies generally. "
We thank Reviewer #2 for this careful reading and for highlighting the central claim of our work. We fully agree that the classification of scattering into Born and unitary limits is an idealization, and that real materials likely fall somewhere in between. Our intention is not to assert that the system lies strictly in one limit or the other, but rather to interpret the observed trends in terms of a crossover from unitary-like to Born-like behavior. To our knowledge, this kind of crossover has never been reported experimentally.
In the absence of a complete theoretical framework for extended or granular disorder in this system, we rely on the experimental signatures—namely, the simultaneous suppression of Tc and the reduction of the residual density of states (DOS) at low disorder—to argue that the effective scattering behavior shifts toward the Born limit as disorder increases. This is in contrast with prior studies in which the residual DOS rapidly increases with small amount of disorder, as expected in the unitary limit. Thus we believe that this new experimental observation deserves publication without sophisticated theories, and such theory should be kept for future work by theorists.
We believe this unusual decrease of residual DOS at low scattering is difficult to reconcile with a purely unitary or even intermediate scattering picture, unless one considers the possibility of competing scattering mechanisms and the influence of the granular nature of superconductivity in this material. We have clarified this point in the revised manuscript and framed our conclusions more cautiously to reflect the complexity of the situation.

"The dependence on the residual DOS decreases with initial cooling rate. This is the most striking result in the article. The authors’ explanation for it is not clearly articulated, at least to this reader. My understanding of the effect of dilute strong (resonant) scatterers (the initial state, slow cooling case) in a d-wave SC is to introduce localized states extending out from the defect location like 4 arms (due to the gap nodal structure).
Is this the right starting point? The authors don’t clearly state it. "
We thank the reviewer for pointing out what we also believe is the most striking result of our work—the initial decrease in residual DOS with increasing disorder.
We presume that “localized states extending out from the defect location like four arms.” refers to real-space picture of impurity-induced states near E = 0 for a d-wave state [Such as Fig.1 of Zhu et al., PRB 67, 094508 (2003)]. Although this work does not focus on the real-space configuration of the impurity-induced states, our answer to Reviewer’s question is basically “yes”. But notice that our system is rather quasi-one-dimensional, thus real-space pattern should be preferentially oriented along the a axis. We added one sentence to the main text explaining possible real-space pattern of the impurity-induced state in the last paragraph of Sec. IV B, together with the newly added reference (Zhu et al. mentioned above).
Irrespective of the real-space pattern, our understanding is instead based on Fig. 3 of Sun and Maki [Phys. Rev. B 51, 6059 (1995)], which illustrates how strong (unitary-limit) scatterers lead to a smearing of the density of states near the Fermi level. This smearing results in a higher residual DOS even for dilute concentrations of such scatterers.
Our central point is that, at low disorder (slow cooling), the system appears to show signatures of unitary-like scattering, while at higher disorder in the ClO4 layer (faster cooling), the behavior becomes more consistent with Born-like scattering, where residual DOS increases more gradually. We have revised the manuscript to explain this proposed crossover more clearly and to better justify our interpretation based on our experimental results.
We hope that the revised discussion now more effectively conveys our reasoning.

"Of course, increasing the number of resonant scatterers would only increase the ratio \gamma_0/\gamma_N (defined in the article as the residual DOS relative to normal state DOS). Here, the point is: what about adding anion-orderered domain walls with increasing density? In order for the ratio \gamma_0/\gamma_N to decrease, I must assume the number of localized states also decreases. The authors have discussed the geometry of their model by introducing the Born scatterers as extended domain walls—a configuration not previously addressed to my knowledge. If they made an argument as to why this reduces the number of localized states, then I missed it."
First, let us note that theories involving scattering from extended defects such as domain boundaries, which are inherently more complex, are, to the best of our knowledge, not currently available. We fully agree that a comprehensive theory accounting for extended scatterers would be necessary to quantitatively capture all effects in future. But such theoretical development is out of the scope of this experimental paper.
In a given sample, the density of chemical impurities is fixed (by the sample fabrication). However, the density of non-magnetic borders of ordered domains increases with the cooling rate. Our interpretation of our experimental results is the following:
• At low cooling rates, the ordered domains are large, encompassing one of several chemical impurities which act as strong scatterers. As a result, each domain has a non-zero DOS.
• When the cooling rate is increased, the domains become smaller (checker board size in Fig. 5). This means they each contain less strong scatterers than at low cooling rates. As a consequence, each domain has a lower DOS. BUT, domain walls have become more frequent, and weak scattering takes over.
• We can imagine a real-space picture as follows: In the initial state, there are “star-like” chemical-impurity-induced states, as Reviewer #2 expected. However, “arms” of the initial impurity-induced states starts to be cut by the ClO4 domain walls as the cooling rate increases. During this process the impurity-induced states near E = 0 is somehow pushed up to higher energies by the Born domain wall scatterings. Nevertheless, this real-space picture is too speculative and should probably be left for future theoretical studies.
This is what we have tried to illustrate in Fig 5. In the revised version of the manuscript, we have tried to be more explicit.
We hope that these replies resolve Reviewer #2’s concerns. Thanks to the comments, the manuscript is substantially improved by the corresponding revisions. We thus believe that the manuscript is worth for consideration for publication in Sci Post.

---

## Round 2 · Referee Report · Anonymous (Referee 3) · 2025-10-19

The referee discloses that the following generative AI tools have been used in the preparation of this report:
This review contains text that has been linguistically refined using generative AI tools (e.g., for grammar and style correction). All scientific content, interpretations, and opinions are solely my own.
Strengths
- This study systematically controls disorder in superconductivity via the cooling rate, enabling a detailed investigation of nonmagnetic scattering effects in the superconducting state through specific heat measurements.
- The presented experimental work is of very high quality.
- This study provides an important experimental test in the field of superconductivity, and the possibility of Born-limit scattering is discussed.
Weaknesses
- The specific heat was measured only above 0.4 K and in zero magnetic field, which limits the reliability of extrapolation and the ability to detect low-temperature effects.
- The experimental data presented are not sufficient to serve as definitive evidence for Born-limit scattering.
Report
As Reviewer #1 pointed out, I also have concerns regarding the background subtraction procedure used in the specific heat analysis to evaluate the residual DOS, gamma0. The specific heat measurements are limited to temperatures above 0.4 K in zero magnetic field. For the sample cooled at 16 K/min (Tc ~ 0.6 K), gamma0 is effectively extrapolated from data taken at temperatures above 0.7Tc. Even for the sample with the highest Tc, the extrapolation is performed from T ~ 0.4Tc. This method is questionable when assessed against standard practices in low-temperature specific heat analysis of superconductors. Particularly in the low-temperature regime, qualitative changes in the temperature dependence can easily arise due to multiband effects, magnetic contributions, or nuclear specific heat. Indeed, low-temperature specific heat measurements reported in Ref. [D. Jerome and S. Yonezawa, C. R. Physique 17, 357 (2016)] revealed a 13% discrepancy in entropy between the superconducting and normal states. As described in the introduction, rapid cooling of (TMTSF)2ClO4 has been reported to preserve anion disorder and stabilize the SDW state. Therefore, with increasing the cooling rate, it is not guaranteed that the normal-state C/T remains constant down to the lowest temperatures, raising concerns about the consistency of the entropy balance analysis. In light of these considerations, it is essential to perform specific heat measurements down to sufficiently low temperatures without relying on extrapolation over a wide temperature range in order to reliably evaluate gamma0.
As an alternative, the authors attempt to distinguish between the Born and unitarity limits using the specific heat jump at Tc. However, this method also raises concerns. While the raw data appear to agree with the predictions of the unitarity theory, the authors argue that the Born theory becomes consistent when the superconducting volume fraction is taken into account. The magnitude of the specific heat jump can be influenced by various factors, making it difficult to conclude that the observed behavior definitively confirms the Born limit.
In summary, as noted by Reviewer #1, the experimental data presented are not sufficient to serve as definitive evidence for Born-limit scattering. Nevertheless, the authors have provided new experimental indications, albeit inconclusive, of Born-limit behavior through challenging measurements, which is a noteworthy contribution. This work has the potential to ispire further research, both experimental and theoretical. Therefore, I recommend publication, provided that the authors incorporate the concerns raised here into the manuscript, clearly acknowledge the limitations, and revise the conclusion to reflect a more cautious interpretation.
Requested changes
The manuscript should be revised to moderate the claim regarding Born-limit scattering and to emphasize the need for further investigation to establish it more conclusively.
Recommendation
Ask for minor revision
Strengths
Report
I retain my original numbering, as the authors used #1 twice. 1. This statement is incorrect. In general, the variation of Tc depends on the scattering strength for any (except for an isotropic-gap) superconductor, especially “d-wave” suggested in this manuscript. 2. Acceptable. 3. The response is unclear. Well-developed methods exist, both numerical (t-matrix) and analytical (see, e.g., work by Hirschfeld and co-workers). Their applicability to granular systems is a separate issue, and this applies to any superconductivity theory developed for homogeneous systems. Regarding probes: the London penetration depth measures the superfluid density directly, whereas the heat capacity contains contributions from all types of quasiparticles (including phonons). Consequently, the heat-capacity measurement is much more ambiguous. 4. This point was not about the residual density of states. I do not think the presented data justify the d-wave pairing picture. The literature remains unsettled; for example,\muSR measurements clearly demonstrate a nodeless gap (doi:10.1103/PhysRevLett.110.107005), contradicting the heat-capacity data. Furthermore, the crystal is triclinic and quasi-1D, which rules out d-wave symmetry. One could refer to a “nodal, like d-wave” state, but not specifically to d-wave. The residual DOS in the present manuscript could simply originate from a non-superconducting fraction in a highly inhomogeneous granular system. 5. OK 6. This is the crucial issue for the paper’s conclusions. There is no reliable method for background subtraction in heat capacity measurements. My comment, therefore, stands: there is no objective proof that the simple procedure employed by the authors is valid. 7. The response is unsatisfactory. It is always possible to estimate the dimensionless scattering strength; essentially, one needs only the mean free path and the BCS coherence length. 8. This comment remains unanswered and is again critical for the interpretation of the data. Scattering on point defects is very different from scattering on extended defects.
Given the substantial unresolved issues, I do not consider this manuscript suitable for publication in SciPost Physics.
Recommendation
Reject

---

## Round 2 · Author Response

Response to the reviews of scipost_202505_00013v1: Born-limit scattering and pair-breaking crossover in d-wave superconductivity of (TMTSF)2ClO4
We would like to thank the reviewers for their careful review of our manuscript. Below are our answers to their concerns and the summary of the changes made in the manuscript to address them.
Report #1 Strengths "The manuscript presents a study of the effects of nonmagnetic disorder on the specific heat of unconventional superconductor (TMTSF)₂ClO₄. The authors conclude that the only way to distinguish between Born and unitary scattering limits is to examine the residual density of states, obtained from the specific heat." We thank Reviewer #1 for approving the method at the core of our paper. Weaknesses "There are a number of issues. 1. The manuscript states: “However, while this extension predicts the effect of increased scattering on Tc , it does not provide any information how scattering acts on the superconducting order parameter.” This statement is incorrect. For any superconductor, conventional or unconventional, Tc is always proportional to the magnitude of the order parameter, just with different pre-factors. " We admit that the latter half of this sentence was misleading. We meant that measuring Tc as a function of scattering does not provide any information on whether the scattering is weak or strong. Thus, we have modified this sentence in the revised manuscript so that it is clearer. The text now reads “However, while this extension predicts the effect of increased scattering on Tc, it does not provide any information on whether the scattering is weak or strong”.
-
"The discussion of the density of states, N(E,k) ( DOS) in the introduction should be more specific. Heat capacity includes the integral over the Fermi surface containing N(E,k) of DOS, so it includes all energies. Unitary scattering primarily affects the states at energies close to the Fermi level, whereas Born limit influences the states near the gap edge, see V. G. Kogan, R. Prozorov, and V. Mishra, London penetration depth and pair breaking, Phys. Rev. B 88, 224508 (2013). Therefore DOS is affected quite significantly in both regimes. It is true, however, that in the Born limit, a very significant scattering rate is required to suppress Tc and superfluid density, hence affecting any thermodynamic quantity, such as heat capacity. This is the method the manuscript uses to distinguish between two regimes." We thank Reviewer #1 for the helpful clarification regarding the energy dependence of the density of states N(E, k) and its role in modifying the heat capacity. As Reviewer #1 notes, heat capacity integrates in the energy range of the order of kBT and is therefore a relevant quantity for capturing the effects of disorder and distinguish between unitary and Born scattering regimes. We appreciate Reviewer #1’s recognition that, in the Born limit, a large scattering rate is needed to noticeably impact thermodynamic quantities such as heat capacity — a point that underlies our method for distinguishing between the two regimes. We have added the reference provided by the reviewer and thank him/her for bringing this paper to our attention. In the revised manuscript, we have clarified this point in the introduction to better convey why heat capacity is an appropriate and sensitive probe in this context.
-
"The authors state that “Almost all unconventional superconductors follow…” unitary limit. This may not quite accurate. Historically, this was the sole theory explaining a rapid change of Tc with non-magnetic disorder in high-Tc. Somehow the Authors do not cite this seminal paper, P. J. Hirschfeld and N. Goldenfeld, Effect of strong scattering on the low-temperature penetration depth of a d-wave superconductor, Phys. Rev. B 48, 4219 (1993). Over time, it has been recognized that the scattering potential strength is usually intermediate, see for example (and references therein), K. Cho, M. Kończykowski, S. Teknowijoyo, S. Ghimire, M. A. Tanatar, V. Mishra, and R. Prozorov, Intermediate scattering potential strength in electron-irradiated YBa2Cu3O7−δ from London penetration depth measurements, Phys. Rev. B 105, 14514 (2022). This generalized potential is treated within the well-developed T-matrix approach. It is believed that end-points of the scattering potential (Born and unitary limits) are never realized and the scattering strength is somewhere in between." We thank Reviewer #1 for pointing out this important historical context and for highlighting the relevant references. We have added the seminal work by Hirschfeld and Goldenfeld [Phys. Rev. B 48, 4219 (1993)] as well as the more recent study by Cho et al. [Phys. Rev. B 105, 014514 (2022)] to the revised manuscript. Accordingly, we have modified our original statement to acknowledge that some unconventional superconductors exhibit scattering strengths that are intermediate between the Born and unitary limits, rather than strictly adhering to the unitary regime. We would like to emphasize that much of the insight into scattering regimes has come from London penetration depth studies, whereas our own measurements are thermodynamic (heat capacity). We now emphasize that the Born and unitary limits are useful idealized cases and serve as endpoints of a continuum of scattering strengths. A complete theoretical description that captures the effective scattering potential in effectively granular materials (such as (TMTSF)2ClO4) or intermediate regimes remains, to the best of our knowledge, to be developed. We hope these changes improve the clarity and balance of the discussion.
-
"For a generalized theory of the effects of arbitrary scattering and order parameters, I recommend consulting L. A. Openov's work: Effect of nonmagnetic and magnetic impurities on the specific heat jump in anisotropic superconductors, Phys. Rev. B 69, 224516 (2004). " We thank Reviewer #1 for bringing the work of L. A. Openov [Phys. Rev. B 69, 224516 (2004)] to our attention. We were not previously aware of this reference and have now added it to the manuscript. Openov’s analysis of the specific heat jump in the presence of nonmagnetic and magnetic impurities, particularly in mixed s- and d-wave order parameters, provides valuable context. As far as we understand, this work does not address the residual density of states directly, and provides predictions for the specific heat jump. Our experimental data are consistent with a purely d-wave order parameter, corresponding to the case r=0 in Openov’s model.
-
"Stylistically, it appears the authors use “resonant scattering” and “unitary scattering” interchangeably, which are distinct regimes. The former produces in-gap bound states, while the latter affects DOS smoothly, as explained in the first referenced paper above." We thank Reviewer #1 for pointing out this. We had indeed used “resonant scattering” once in the conclusion. We’ve modified it to “unitarity scattering” to align with Maki et al.’s denomination.
-
"The authors apply an over-simplified background subtraction developed for isotropic metals with isotropic pairing. Phonon modes are sensitive to any anisotropy. More critically, the authors assume there is no electronic contribution above Tc, stating: “after subtracting this background contribution, the phonon contribution was obtained by fitting Cphonon/T = B22T^2+B44T^4 to the data above Tc.” It is likely they used, but neglected to include, the electronic term (which would be a constant) in the text. This procedure has significant uncertainty due to the fitting region being far from T=0, and because the formula is derived for a simple Fermi-liquid model, which does not apply to many unconventional superconductors. Usually, phonons are subtracted by suppressing superconductivity using a magnetic field, though magnetoresistance is a problem. (Another way is to use a non-magnetic analog of the studied compound, but this is inapplicable here.) Given the reliance on the subtraction procedure, I do not know how reliable the electronic part is. This important point should be addressed in detail." First of all, we apology the mistype in the previous manuscript. Actually, we did include the electronic term in the fitting as a standard procedure. The part mentioned by Reviewer #1 should have been written as “after subtracting this background contribution, the phonon contribution was obtained by fitting C/T = gamma + Cphonon/T to the data above Tc, where Cphonon/T is assumed to have the form B22T^2 + B44T^4”. We now modified the 2nd paragraph of Appendix C. I hope this clarifies most of Reviewer’s concern. It is true that suppressing superconductivity by magnetic field is often used to estimate phononic contribution. Nevertheless, it is not straightforward to perform in-field measurements in the present setup using an adiabatic demagnetization refrigerator, where zero-field condition is necessary to cool down the sample. We add a short explanation on this point in the 2nd paragraph of Appendix C. On the influence of the anisotropy of phonon modes, although we agree that our model is a very simple one, but we are not aware of any widely-accepted and reliable model to include such effects to analyze the specific-heat data. We would like to emphasize that the simple model for the phonon contribution fits the normal-state data well irrespective of the cooling rate. We also comment that more complicated models may have risks of overfitting of the data. We thus believe that the simple model is phenomenologically good enough for our purpose. To address Reviewer #1’s concern, we added a short description in the 2nd paragraph of Appendix C.
-
"Another concern is that the manuscript shows only normalized scattering rates, which is obtained from the normalized resistivity Ref.[33], see Fig.2. This is misleading. The authors should derive the absolute values of the scattering rate, \Gamma, and plot it along with the AG-Openov curve for a d-wave superconductor with non-magnetic impurities, which has a specific critical value of \Gamma_c=0.28 when Tc is suppressed to zero. Depending on gap anisotropy, the function Tc(\Gamma) can be anywhere from a scattering-independent constant to the AGO curve." It is true that evaluation of the absolute value of Γc is ideal, but experimentally, accurate quantitative evaluation of Γc is often very difficult. Instead, in many experimental papers of various unconventional superconductors, the ratio Γ/Γc (or just the residual resistivity) is used to compare experimental data with the AG framework. The difficulty in finding Γc is exemplified by the fact that Γc is not universal and depends on the nature of impurities. In the (TMTSF)2ClO4 system, this is corroborated by the following references: Joo et al. [Eur. Phys. J. B 40, 43 (2004)] reports the critical residual resistivity of ρc ≈ 0.185 Ω⋅cm for (TMTSF)2ClO4/ReO4 solid solutions, while for pure (TMTSF)2ClO4, Yonezawa et al. [Phys Rev B 97, 014521 (2018)] reports ρc ≈ 0.24 Ω⋅cm. For the present study, our fitting in Fig.2(c) yields 0.14 Ω⋅cm (this small value may be attributable to the fact that bulk thermodynamic quantity is used in this study, whereas resistivity is used to evaluate Tc in the previous two studies). Thus, Γc, which is proportional to ρc, would differ by 170% among different data sets. Because of this fact, as well as many examples of previous papers on unconventional superconductors, the use of Γ/Γc in the present experimental paper should be justified. To address Reviewer’s concern, we added the value of the critical resistivity in the last paragraph of Sec. IIIA. We emphasize that, besides the difference in the critical scattering, Tc dependence on the scattering is quite well explained by the AG theory for d-wave superconductivity influenced by non-magnetic impurities.
-
"All mentioned theories assume point-like scattering potential. The model suggested in the manuscript involves domain boundaries. Scattering on extended defects is considerably more complex has very different effect on properties." We thank Reviewer #1 for this important observation. Indeed, most standard theoretical treatments, including those we reference, assume point-like impurity scattering. In contrast, the model we consider involves scattering from extended defects such as domain boundaries, which are inherently more complex and can influence quasiparticle properties differently. We fully agree that a comprehensive theory accounting for extended scatterers is still lacking and would be necessary to quantitatively capture all effects. Our intention was to provide a phenomenological framework motivated by the experimental trends, while acknowledging that the microscopic scattering mechanisms—particularly in the granular regime—are likely to differ from those of point-like impurities.
Report "It is evident that considerable effort was put into this work, and the authors did their best to present and analyze the data. However, the significant issues outlined above need to be addressed before considering it for any journal." We sincerely appreciate Reviewer #1’s recognition of the effort invested in this experimental work. We would like to emphasize that our primary focus is on presenting and analyzing new experimental data, with the aim of stimulating further theoretical developments. We also mention that most serious concerns of Reviewer #1 are due to our writing issue and are now fully resolved. Thus, we now believe that the manuscript is ready for consideration of publication.
Report #2 Strengths "The system under study lends itself to study the effect of (variable density, in situ) disorder on the residual density of states simultaneously with the effect on the superconducting transition temperature." We thank Reviewer #2 for recognizing the value of (TMTSF)2ClO4, which disorder can be fine-tuned and thus serve as a model system for studying the scattering in d-wave superconductors. Weaknesses 1. "The argument given for explaining the drop in apparent DOS on defect density is unclear, and Fig. 3 only marginally helps in the explanation. " We assume Reviewer #2 refers to Fig.5? The main idea we are trying to convey is that, as the cooling rate changes, the density of chemical impurities is fixed, but the density of non-magnetic borders of ordered domains increases. This is what we have tried to illustrate in Fig 5. In the revised version of the manuscript, we have tried to be more explicit. 2. "The starting point of "strong scatterers" is not clearly defined (see the Report below)." See response below. Report "The manuscript entitled “Born-limit scattering and pair-breaking crossover in d-wave superconductivity of (TMTSF)2ClO4” (Yano, et al.) documents simultaneously the rise of the residual DOS and the depression of the SC critical temperature Tc with increased disorder. The residual DOS and Tc are determined from ac calorimetry measurements. The disorder is indirectly controlled by the rate of cooling through an anion ordering transition that takes place at higher temperature. The authors explain the Fermi surface is indirectly affected as a result since there is a period doubling of the unit cell along the b-direction with slow cooling. From the variation of the residual DOS with cooling rate, the authors conclude that the defects introduced by increasing the cooling rate are Born-limit scatterers, and consequently very different than defects studied previously in unconventional superconductors. The claim is a strong one that requires explicit defense. My understanding is that neither limit applies generally. " We thank Reviewer #2 for this careful reading and for highlighting the central claim of our work. We fully agree that the classification of scattering into Born and unitary limits is an idealization, and that real materials likely fall somewhere in between. Our intention is not to assert that the system lies strictly in one limit or the other, but rather to interpret the observed trends in terms of a crossover from unitary-like to Born-like behavior. To our knowledge, this kind of crossover has never been reported experimentally. In the absence of a complete theoretical framework for extended or granular disorder in this system, we rely on the experimental signatures—namely, the simultaneous suppression of Tc and the reduction of the residual density of states (DOS) at low disorder—to argue that the effective scattering behavior shifts toward the Born limit as disorder increases. This is in contrast with prior studies in which the residual DOS rapidly increases with small amount of disorder, as expected in the unitary limit. Thus we believe that this new experimental observation deserves publication without sophisticated theories, and such theory should be kept for future work by theorists. We believe this unusual decrease of residual DOS at low scattering is difficult to reconcile with a purely unitary or even intermediate scattering picture, unless one considers the possibility of competing scattering mechanisms and the influence of the granular nature of superconductivity in this material. We have clarified this point in the revised manuscript and framed our conclusions more cautiously to reflect the complexity of the situation.
"The dependence on the residual DOS decreases with initial cooling rate. This is the most striking result in the article. The authors’ explanation for it is not clearly articulated, at least to this reader. My understanding of the effect of dilute strong (resonant) scatterers (the initial state, slow cooling case) in a d-wave SC is to introduce localized states extending out from the defect location like 4 arms (due to the gap nodal structure). Is this the right starting point? The authors don’t clearly state it. " We thank the reviewer for pointing out what we also believe is the most striking result of our work—the initial decrease in residual DOS with increasing disorder. We presume that “localized states extending out from the defect location like four arms.” refers to real-space picture of impurity-induced states near E = 0 for a d-wave state [Such as Fig.1 of Zhu et al., PRB 67, 094508 (2003)]. Although this work does not focus on the real-space configuration of the impurity-induced states, our answer to Reviewer’s question is basically “yes”. But notice that our system is rather quasi-one-dimensional, thus real-space pattern should be preferentially oriented along the a axis. We added one sentence to the main text explaining possible real-space pattern of the impurity-induced state in the last paragraph of Sec. IV B, together with the newly added reference (Zhu et al. mentioned above). Irrespective of the real-space pattern, our understanding is instead based on Fig. 3 of Sun and Maki [Phys. Rev. B 51, 6059 (1995)], which illustrates how strong (unitary-limit) scatterers lead to a smearing of the density of states near the Fermi level. This smearing results in a higher residual DOS even for dilute concentrations of such scatterers. Our central point is that, at low disorder (slow cooling), the system appears to show signatures of unitary-like scattering, while at higher disorder in the ClO4 layer (faster cooling), the behavior becomes more consistent with Born-like scattering, where residual DOS increases more gradually. We have revised the manuscript to explain this proposed crossover more clearly and to better justify our interpretation based on our experimental results. We hope that the revised discussion now more effectively conveys our reasoning.
"Of course, increasing the number of resonant scatterers would only increase the ratio \gamma_0/\gamma_N (defined in the article as the residual DOS relative to normal state DOS). Here, the point is: what about adding anion-orderered domain walls with increasing density? In order for the ratio \gamma_0/\gamma_N to decrease, I must assume the number of localized states also decreases. The authors have discussed the geometry of their model by introducing the Born scatterers as extended domain walls—a configuration not previously addressed to my knowledge. If they made an argument as to why this reduces the number of localized states, then I missed it." First, let us note that theories involving scattering from extended defects such as domain boundaries, which are inherently more complex, are, to the best of our knowledge, not currently available. We fully agree that a comprehensive theory accounting for extended scatterers would be necessary to quantitatively capture all effects in future. But such theoretical development is out of the scope of this experimental paper. In a given sample, the density of chemical impurities is fixed (by the sample fabrication). However, the density of non-magnetic borders of ordered domains increases with the cooling rate. Our interpretation of our experimental results is the following: • At low cooling rates, the ordered domains are large, encompassing one of several chemical impurities which act as strong scatterers. As a result, each domain has a non-zero DOS. • When the cooling rate is increased, the domains become smaller (checker board size in Fig. 5). This means they each contain less strong scatterers than at low cooling rates. As a consequence, each domain has a lower DOS. BUT, domain walls have become more frequent, and weak scattering takes over. • We can imagine a real-space picture as follows: In the initial state, there are “star-like” chemical-impurity-induced states, as Reviewer #2 expected. However, “arms” of the initial impurity-induced states starts to be cut by the ClO4 domain walls as the cooling rate increases. During this process the impurity-induced states near E = 0 is somehow pushed up to higher energies by the Born domain wall scatterings. Nevertheless, this real-space picture is too speculative and should probably be left for future theoretical studies. This is what we have tried to illustrate in Fig 5. In the revised version of the manuscript, we have tried to be more explicit. We hope that these replies resolve Reviewer #2’s concerns. Thanks to the comments, the manuscript is substantially improved by the corresponding revisions. We thus believe that the manuscript is worth for consideration for publication in Sci Post.

---

## Round 2 · List of Changes

- We have clarified the manuscript in a number of places. These are detailed in the "author comments" and major changes are highlighted in red in the revised version of the manuscript.
- The main change is that we have modified our original statement to acknowledge that some unconventional superconductors exhibit scattering strengths that are intermediate between the Born and unitary limits, rather than strictly adhering to the unitary regime.
- We corrected and clarified the manner in which the electronic specific heat was extracted.

---

## Editorial Decision

unknown